# Salivary Immunoglobulin Gamma-3 Chain C Is a Promising Noninvasive Biomarker for Systemic Lupus Erythematosus

**DOI:** 10.3390/ijms22031374

**Published:** 2021-01-29

**Authors:** Ju-Yang Jung, Jin-Young Nam, Keun-Sil Ryu, In-Ok Son, Joo-Ho Shin, Wook-Young Baek, Hyoun-Ah Kim, Chang-Hee Suh

**Affiliations:** 1Department of Rheumatology, Ajou University School of Medicine, Suwon 16499, Korea; serinne20@hanmail.net (J.-Y.J.); 500097@aumc.ac.kr (J.-Y.N.); ks-blue@hanmail.net (K.-S.R.); 119iyla@naver.com (I.-O.S.); nakhada@naver.com (W.-Y.B.); kiborojh@gmail.com (H.-A.K.); 2Division of Pharmacology, Department of Molecular Cell Biology, Sungkyunkwan University School of Medicine, Suwon 16419, Korea; arikato83@naver.com

**Keywords:** systemic lupus erythematosus, proteomics, salivary proteins and peptides, biomarker

## Abstract

We aimed to characterize the salivary protein components and identify biomarkers in patients with systemic lupus erythematosus (SLE). A proteomic analysis using two-dimensional gel electrophoresis and mass spectrometry was performed to determine the alterations of salivary proteins between patients with SLE and healthy controls, and the concentrations of the candidate proteins were measured through Western blot analysis and the enzyme-linked immunosorbent assay. The 10 differentially expressed protein spots were immunoglobulin gamma-3 chain C region (IGHG3), immunoglobulin alpha-1 chain C region, protein S100A8, lactoferrin, leukemia-associated protein 7, and 8-oxoguanine DNA glycosylase. The patients with SLE exhibited enhanced salivary IGHG3 (3.9 ± 2.15 pg/mL) and lactoferrin (4.7 ± 1.8 pg/mL) levels compared to patients with rheumatoid arthritis (1.8 ± 1.01 pg/mL and 3.2 ± 1.6 pg/mL, respectively; *p* < 0.001 for both) or healthy controls (2.2 ± 1.64 pg/mL and 2.2 ± 1.7 pg/mL, respectively; *p* < 0.001 for both). The salivary IGHG3 levels correlated with the erythrocyte sedimentation rate (*r* = 0.26, *p* = 0.01), anti-double-stranded DNA (dsDNA) antibody levels (*r* = 0.25, *p* = 0.01), and nephritis (*r* = 0.28, *p* = 0.01). The proteomic analysis revealed that the salivary IGHG3 levels were associated with SLE and lupus disease activity, suggesting that salivary IGHG3 may be a promising noninvasive biomarker for SLE.

## 1. Introduction

Systemic lupus erythematosus (SLE) is a heterogeneous autoimmune disease characterized by the production of pathogenic autoantibodies and an aberrant inflammatory response leading to diverse clinical manifestations [1]. The disease status of SLE, including clinical manifestations and disease activity, varies with the progression of the disease. However, there is a lack of indicators to represent changes in the SLE disease status. Currently, anti-double-stranded DNA (dsDNA) antibody and complement protein levels are used as the markers for diagnosis or monitoring of SLE [2,3]. The anti-dsDNA antibodies target the intracellular DNA and induce apoptosis. The anti-dsDNA antibodies cross-react with the α-actin present in the glomeruli of patients with SLE exhibiting renal disease [4]. Although it has been known that high titers of anti-dsDNA antibody reflect the disease activity of SLE, contradictory results have been reported by studies investigating the association of anti-dsDNA antibody with disease flares within a few weeks or months [5,6,7]. The activation of a complement system leads to the consumption or depletion of complement proteins during SLE disease flares [8]. Moreover, deficiencies of the complement proteins involved in the classical complement pathway are reported to confer susceptibility to SLE. Clinically, the low levels of complements 3 and 4 have been used as a diagnostic or a disease activity marker in SLE [1]. However, complement proteins are not a reliable indicator of active SLE status sometimes, as their concentrations vary widely, and the complement protein level does not indicate the consumption in the tissue or the presence of an anticomplement autoantibody [9]. The changes in the expression levels of these biomarkers are often nonspecific and, thus, are not a reliable indicator of the progression of SLE. Therefore, several studies are focused on identifying new biomarkers for SLE. Several candidates, such as cytokines, immune cells, autoantibodies, or genetic markers, have been identified as potential biomarkers for SLE.

Proteomics is used to detect the protein or peptide in the body fluids or tissues. Two-dimensional gel electrophoresis (2DE) with mass spectrometry (MS) is a reliable method for the identification and characterization of protein constituents in the tissues or body fluids, including blood, urine, or saliva. The 2DE with MS method can detect protein constituents with high sensitivity and specificity [10,11]. The 2DE with MS method has been used to identify novel biomarkers for the diagnosis or disease monitoring of rheumatic diseases [12].

Saliva is a body fluid that can be collected repeatedly using a noninvasive and risk-free procedure [13]. Salivary proteins are derived from the salivary glands and blood. As the composition of saliva is similar to that of blood, saliva is used to identify the disease biomarkers for not only oral disorders but, also, for systemic diseases [14,15,16]. Some studies have suggested that salivary cytokines secreted by B cells are potential biomarkers for Sjogren’s syndrome (SS) [17]. Salivary biomarkers are promising candidates for the diagnosis of SS, which is an autoimmune disorder involving the salivary glands, and shares several common characteristics with SLE [18,19]. A proteomic analysis revealed a differential salivary protein composition between the primary and secondary SS [20]. The levels of salivary α-enolase, β-2 microglobulin, and immunoglobulin kappa light chain are different between patients with SS and those with other autoimmune diseases exhibiting sicca symptoms.

Currently, there are limited studies that have analyzed the saliva samples of patients with SLE. In this study, we analyzed the compositions and concentrations of salivary proteins in patients with SLE. The clinical relevance of the differentially expressed proteins was analyzed with respect to disease characteristics in patients with SLE.

## 2. Results

### 2.1. Salivary Protein Identification

The 2DE analysis revealed a differential salivary protein expression pattern between patients with SLE and healthy controls (HCs) (Figure 1). The protein identity, fold change, and peptide sequence between the two groups were determined by a liquid chromatography tandem mass spectrometry (LC-MS) analysis and quantitative protein profiling. Among the 10 spots exhibiting fold change values ≥ 2, two spots were identified as alpha-amylases: protein S100A8 and 8-oxoguanine DNA glycosylase (OGG1) (Table 1). The other spots were identified as immunoglobulin gamma-3 (IgG3) chain C region (IGHG3), immunoglobulin alpha-1 chain C region (IGHA1), lactotransferrin (or lactoferrin), and leukemia-associated protein 7.

### 2.2. Expressions of Salivary Proteins in Patients with SLE

A Western blot analysis was performed on 12 age-matched saliva samples of patients with SLE, 8 saliva samples of patients with rheumatoid arthritis (RA), and 8 saliva samples of HCs (Figure 2). Among the patients with SLE, the disease duration was 6.6 ± 5.7 years, the systemic lupus erythematosus disease activity index (SLEDAI) was 8.5 ± 4.6, the complement 3 level was 59.3 ± 25.8 mg/dL, the complement 4 (C4) level was 9.4 ± 4.5 mg/dL, and the erythrocyte sedimentation rate (ESR) was 31.4 ± 24.5 mm/h. In addition, 10 patients tested positive for the anti-dsDNA antibody, 5 patients had mucocutaneous symptoms, 5 patients had arthritis, 6 patients had active lupus nephritis (LN), and 5 patients had hematologic symptoms. Among the eight patients with rheumatoid arthritis (RA), the mean disease duration was 5.8 ± 7.6 years, and the mean disease activity score-28 (DAS28) was 2.49 ± 1.23.

The optical densities (ODs) of salivary IGHG3 were 0.91 ± 0.1 in patients with SLE, 0.86 ± 0.09 in patients with RA (vs. SLE, *p* = 0.28), and 0.77 ± 0.06 in HCs (vs. SLE, *p* = 0.004). The ODs of salivary IGHA1 were 0.85 ± 0.15 in patients with SLE, 0.81 ± 0.12 in patients with RA (vs. SLE, *p* = 0.4), and 0.74 ± 0.06 in HCs (vs. SLE, *p* = 0.25). The ODs of salivary lactoferrin were 0.9 ± 0.08 in patients with SLE, 0.85 ± 0.05 in patients with RA (vs. SLE, *p* = 0.19), and 0.77 ± 0.06 in HCs (vs. SLE, *p* = 0.001). The ODs of salivary OGG1 were 0.9 ± 0.09 in patients with SLE, 0.87 ± 0.08 in patients with RA (vs. SLE, *p* = 0.64), and 0.87 ± 0.1 in HCs (vs. SLE, *p* = 0.44). The ODs of salivary S100A8 were 0.91 ± 0.11 in patients with SLE, 0.86 ± 0.09 in patients with RA (vs. SLE, *p* = 0.32), and 0.69 ± 0.09 (vs. SLE, *p* = 0.11) in HCs.

The repeated Western blot analysis revealed that the patients with SLE exhibited an enhanced expression of salivary IGHG3 and lactoferrin compared to HCs (*p* < 0.05). However, the expression levels of salivary IGHA1, S100A8, and OGG1 in patients with SLE were not significantly different when compared to those in patients with RA and HCs.

### 2.3. Expression of Salivary IGHG3 and Lactoferrin in SLE

The levels of salivary IGHG3 and lactoferrin were measured using an enzyme-linked immunosorbent assay (ELISA) (Figure 3). The levels of salivary IGHG3 were elevated (3.9 ± 2.15 pg/mL) in patients with SLE compared to those in patients with RA (1.8 ± 1.01 pg/mL, *p* < 0.001) and HCs (2.2 ± 1.64 pg/mL, *p* < 0.001). The levels of salivary lactoferrin in patients with SLE were elevated (4.7 ± 1.8 pg/mL) compared to those in RA (3.2 ± 1.6 pg/mL, *p* < 0.001) and HCs (2.2 ± 1.7 pg/mL, *p* < 0.001).

The receiver operating characteristic (ROC) curve analysis of salivary IGHG3 and lactoferrin revealed that the areas under the curve (AUC) were 0.75 (95% confidence interval (CI) 0.68–0.83) and 0.84 (95% CI 0.78–0.9), respectively (Figure 4). The sensitivity and specificity of salivary IGHG3 were 78% and 64.6%, respectively, with a cut-off value of 2.26 pg/mL for the diagnosis of SLE. The sensitivity and specificity of salivary lactoferrin were 91.9% and 60.6%, respectively, with a cut-off value of 4.36 pg/mL for the diagnosis of SLE.

### 2.4. Correlation of Salivary Proteins and Clinical Features in SLE

The concentrations of salivary IGHG3 correlated with the ESR (*r* = 0.26, *p* = 0.01), anti-dsDNA antibody levels (*r* = 0.25, *p* = 0.01), nephritis (*r* = 0.28, *p* = 0.01), and use of immunosuppressants (*r* = 21, *p* = 0.04), whereas the levels of salivary lactoferrin were not correlated (Table 2). Additionally, the concentration of salivary IGHG3 in patients with LN (4.66 ± 1.87 pg/mL) was significantly higher than that in patients without LN (3.57 ± 2.2 pg/mL, *p* = 0.006) (Figure 5).

## 3. Discussion

The 2DE with MS proteomic analysis of saliva revealed that the densities of the ten spots were significantly different between patients with SLE and HCs. These spots were identified as IGHG3, IGHA1, protein S100, lactoferrin, OGG1, and leukemia-associated protein 7. The immunoblotting analysis revealed that the expression levels of salivary IGHG3 and lactoferrin in patients with SLE were significantly higher than those in patients with RA and HCs. Additionally, the salivary IGHG3 levels correlated with the ESR, anti-dsDNA antibody, LN, and immunosuppresssants in patients with SLE.

The immunoglobulin gamma 3 heavy chain constant region binds to the Fcγ receptor (FcγR) of neutrophilic granulocytes and macrophages. The aberrant expression of FcγR for IgG was observed in patients with SLE, and FcγR is involved in antigen presentation, the maturation of dendritic cells, and plasma cell survival in SLE [21]. The pathological roles of IgG, a subunit of autoantibodies, have been known in SLE, but IgG3 or IGHG3 has not been investigated well [22,23]. In a study on patients with autoimmune hemolytic anemia, the levels of IGHG3 in red blood cells (RBCs) were associated with the frequency of an RBC transfusion after diagnosis [24]. The enhanced expression of IGHG3, which acts as an immunoprotein in RBC, might lead to hemolysis via an autoimmune reaction. Blood cells, including RBC and platelets, are destructed by the autoimmune response, and diverse autoantibodies comprising IgG have been revealed to induce cytopenias in SLE [25,26]. However, salivary IgG3 levels are not correlated with any blood cells or hematologic disease. The levels of anti-nucleosomes of the IgG1 and IgG3 subclasses were higher in patients with active SLE than in patients with other connective tissue diseases [27]. Most subclasses of IgG, including IgG2, IgG3, and IgG4, were highly expressed in the skin tissues of patients with SLE [28]. Moreover, the levels of IgG4, but not those of IgG3, were higher in the serum of patients with SLE than in those of patients with RA or HCs, and the ratio of IgG4/IgG1-4 was negatively correlated with a complement deposition [29]. In addition, a study investigating the B-cell receptor repertoire in immune-mediated diseases, including SLE, showed that the levels of the IgA1/2 and IgG1/2 isotypes, but not those of the IgG3 isotype, were significantly higher in patients with SLE compared to the HCs [30]. In the current study, we also assessed the levels of serum IgG3 in a small population among the groups whose salivary IgG3 concentrations were measured by ELISA. We found that the serum IgG3 levels did not increase as the salivary IgG3 levels in patients with SLE compared to HCs (data not shown). Taken together, the levels of salivary IgG3, but not those of serum IgG3, were increased in patients with SLE; however, the underlying mechanisms should be investigated further. In addition, the levels of salivary IGHG3 were correlated with disease activity markers, such as the ESR and anti-dsDNA antibody levels, which were more elevated in patients with LN than in those without. Therefore, salivary IGHG3 might be a promising noninvasive biomarker for SLE to replace invasive blood tests.

Lactoferrin, also known as lactotransferrin, is a multifunctional glycoprotein that belongs to the transferrin family. Lactoferrin expression is detected in the mucosal secretions and secondary granules of polymorphonuclear leukocytes [31]. Lactoferrin not only plays an important role in protection against microorganisms, but it is also involved in immunomodulation, inflammation, and anticancer activity through its interactions with the host immune system [32,33]. Lactoferrin-specific IgG autoantibodies were detected in the serum of patients with SLE or RA [34]. The release of surface-expressed lactoferrin from the polymorphonuclear neutrophils modulates the cytokine production in T-helper cell type 1 (Th1). The decreased expression of lactoferrin in patients with SLE is associated with abnormal Th1/Th2 production [35]. Lactoferrin-containing immune complexes induce the production of proinflammatory cytokines in the monocytes and monocyte-derived macrophages and can promote the proinflammatory M1-like phenotype of human macrophages [36,37]. In the present study, while the lactoferrin levels were not measured in the tissue or other body fluids of patients with SLE, the levels of salivary lactoferrin were elevated in patients with SLE compared to the HCs and patients with RA, and the ROC curve suggested its ability to differentiate SLE. The mechanisms underlying the increased levels of salivary lactoferrin in patients with SLE need to be investigated in future studies, and their utility as biomarkers should be assessed in a large number of patients with SLE.

There are a few limitations to our study. First, it was difficult to establish statistically significant correlations between salivary protein concentrations and disease activity or manifestations of SLE, as most of the patients were well-controlled by the standard management strategies of SLE and had mild disease activities. Furthermore, in 2D proteomics, the overexpression of certain molecules in a few patients might have led to the differences in spot intensities due to the pooling of samples from 11 patients and HCs.

This study revealed differential salivary protein compositions in patients with SLE and RA and the HCs. The levels of salivary IGHG3 and lactoferrin in patients with SLE were significantly higher than those in patients with RA and the HCs. The concentrations of salivary IGHG3 and lactoferrin may be used as differential biomarkers for diagnosing SLE. Additionally, salivary IGHG3 may be a noninvasive biomarker of disease activity because that level correlated with the ESR and anti-dsDNA antibody and were higher in patients with LN.

## 4. Materials and Methods

### 4.1. Study Participants

The analysis was performed in two steps. In the first step, the differential salivary protein compositions between patients with SLE and the HCs were analyzed by 2DE with MS. In the next step, the differentially expressed proteins identified in the 2DE with MS analysis were validated by Western blotting and ELISA.

The participants were divided into three groups. The first group included 11 patients with SLE and 11 HCs, whose salivary samples (the samples were pooled for each group) were subjected to a 2DE with MS proteomic analysis. The second group included 12 patients with SLE, 8 patients with RA, and 8 HCs, whose salivary samples were subjected to a Western blot analysis. The characteristics of the two groups are presented in the Appendix A. The third group included 94 patients with SLE, 57 patients with RA, and 62 HCs (Table 3). The concentration of proteins in the saliva samples of HCs and patients with SLE or RA was validated by ELISA.

All enrolled patients with SLE met the revised American College of Rheumatology classification criteria [38]. Additionally, age-matched and sex-matched patients with RA and HCs who had no history of autoimmune or inflammatory disorders were enrolled in the study. This study was conducted according to the Declaration of Helsinki and Good Clinical Practice guidelines. Informed consent was obtained from all participants enrolled in this study. The study protocol was approved by the Institutional Review Board of Ajou University Hospital (BMR-SMP-13-199, 15 May 2014).

Information on the medical history and clinical manifestations was collected from a chart review and blood test results, such as a complete blood count, ESR, antinuclear antibody levels, C3 and C4 levels, and anti-dsDNA antibody levels. Patients with RA were enrolled as a disease control to analyze the differential expression of specific proteins between SLE and RA, which are both chronic autoimmune diseases.

The basic characteristics of the second participant group are shown in Table 3. The mean ages of patients with SLE, patients with RA, and HCs were 39.8 ± 9.8, 41 ± 7.9, and 39.5 ± 6.9 years, respectively, which were not significantly different between the three groups. Among the patients with SLE, 41 patients (43.6%) tested positive for the anti-dsDNA antibody, 28 patients (29.8%) had mucocutaneous symptoms, 31 patients (33.0%) had arthritis, and 29 patients (30.9%) had nephritis. The mean SLEDAI was 3.8 ± 4.2. Among the 57 patients with RA, 45 patients (77.6%) tested positive for rheumatoid factors, and their mean DAS28 was 3.3 ± 1.15. Regarding treatment information, 69 patients (73.4%) were undergoing glucocorticoid therapy, the mean dose of glucocorticoids was 3.1 ± 3.7 mg (prednisolone-equivalent), and 35 patients (37.2%) were undergoing treatment with immunosuppressive agents.

### 4.2. Saliva Sample Collection

As salivary proteins levels exhibited diurnal variations, and the saliva samples were collected from all participants between 9:00 and 11:00 a.m. The subjects were not allowed to eat, drink, smoke, or perform oral hygiene procedures for at least 1 h prior to the sample collection. The saliva samples were collected for 5 min after the subjects rinsed their mouths with water, as previously described [39]. Saliva secretion was not stimulated in the study subjects. The subjects were asked to keep their mouths closed and expectorate the saliva into a tube once per minute. Each saliva sample was immediately treated with the protease inhibitors to preserve the integrity of the protein constituents. The saliva samples were centrifuged at 9425× *g* for 15 min at 4 °C and stored at −20 °C until further analysis.

### 4.3. Two-Dimensional Gel Electrophoresis

The samples from 11 patients with SLE or 11 HCs were pooled equally to avoid intraclass variations that were detected between the patients in the 2DE analyses. A 1-mL aliquot of the sample was concentrated 10 times using Amicon 3-K centrifugal filters (Sigma, St. Louis, MO, USA) by centrifuging at 14,000× *g* at 4 °C for 20 min. The proteins in the salivary samples were precipitated using 500 μL of a trichloroacetic acid/acetone (90%; *v/v*)–dithiothreitol mixture overnight at −20 °C. The samples were centrifuged at 9425× *g* at 10 °C for 10 min. The supernatant was collected, and the samples were pretreated with 250 μL of rehydration buffer. Next, the samples were centrifuged at 9425× *g* at 10 °C for 10 min to remove any insoluble materials. The protein concentrations of the samples were estimated using the Bradford protein assay (Bio-Rad, Hercules, CA, USA).

### 4.4. Liquid Chromatography Tandem Mass Spectrometry

Following the 2DE proteomic analysis of the saliva samples of 11 patients with SLE and 11 HCs, the proteins separated into numerous spots with different concentrations. The proteins in 10 spots were subjected to LC-MS to analyze the proteins with high specificity [40]. The gel pieces containing the protein spots were destained, reduced, alkylated, and digested with modified sequencing grade trypsin (Sigma, St. Louis, MO, USA), as previously described [41]. Peptide mixtures were lyophilized and stored at −80 °C for further LC-MS analysis.

Each sample was resuspended in 0.1% trifluoroacetic acid and injected into a Zorbox 300SB-C18 75 μm × 15 cm column (Agilent, Santa Clara, CA, USA) via the trap column. The peptides were separated in an acetonitrile gradient at a flow rate of 200 nL/min in an UltiMate 3000 nano high performance liquid chromatography (HPLC) system (Dionex, Sunnyvale, CA, USA). The peptides were then applied online to an LTQ ion trap mass spectrometer (Thermo Fisher, Waltham, MA, USA). The mobile phase gradient was initiated with an increase from 5% to 40% buffer within 110 min. Next, the gradient was increased to an 80% buffer in 1 min. The gradient was maintained at the isocratic conditions of the 80% buffer for 15 min. The main working liquid junction electrospray ionization source parameters were as follows: ion spray voltage, 1.6 kV; capillary voltage, 24 V; and capillary temperature, 200 °C. The spectra were obtained in full scan mode using the dynamic exclusion criteria. LC-MS runs were analyzed using DeCyder MS (version 2.0; GE Healthcare, Uppsala, Sweden) software [42]. The peptide peaks were detected with an average peak width of 1 min and matched with a mass accuracy of at least 0.6 Da and a maximum time window of 4 min. The abundances of individual peptides in their respective gradient fractions were calculated by peak integration.

The data were manually examined, and the overlapping peaks were discarded. The threshold level for differentially expressed proteins was defined as at least a 2-fold increase or decrease in spot intensity that was statistically significant. The MS spectra of the peptide peaks were searched against the UniProt Human database using Mascot version 2.3 (Matrix Science, London, UK). For quantitative protein profiling, only the proteins identified by multiple peptides with significant MASCOT scores (*p* < 0.05) were considered.

### 4.5. Western Blot Analysis

IGHG3, IGHA1, S100A8, lactoferrin, and OGG1 were analyzed by Western blotting using a rabbit anti-human IGHG3 polyclonal (MBS248789; MyBioSource, San Diego, CA, USA), rabbit anti-human IGHA1 polyclonal (MBS9206028; MyBioSource), rabbit anti-human rat S100A8 polyclonal (MBS127619; MyBioSource), mouse anti-human lactoferrin monoclonal (ab10110; Abcam, Cambridge, UK), and rabbit anti-human OGG1 polyclonal (NB100-106; Novus Biologicals, Centennial, CO, USA) antibodies, respectively. The proteins were subjected to polyacrylamide gel electrophoresis using 10% (for IGHG3) or 15% (for IGHA1, S100/A8, lactoferrin, and OGG1) gel. The resolved proteins were transferred to polyvinylidene fluoride membranes. The membranes were incubated with secondary antibodies (goat anti-rabbit antibody A120-101P for IGHG3, IGHA1, S100A8, and OGG1 and goat anti-mouse antibody for lactoferrin; Bethyl Laboratories, Montgomery, TX, USA) diluted at 1:10,000 (IGHG3 and S100/A8) and 1:2000 (IGHA1, lactoferrin, and OGG). All analyses were performed in triplicate. The protein concentration was determined by measuring the optical density of the specific immunoreactive bands using Image J software (NIH, Bethesda, MD, USA).

### 4.6. Enzyme-Linked Immunosorbent Assay

The levels of salivary IGHG3 and lactoferrin were measured in patients with SLE or RA and HCs by ELISA using the human IGHG3 ELISA kit (ab137981; Abcam) and human lactoferrin ELISA kit (ab108882; Abcam), respectively, following the manufacturer’s instructions. All measurements were performed in duplicate.

### 4.7. Statistical Analysis

The two-sample Wilcoxon rank-sum (Mann–Whitney) test was used to compare the differences in the expressions of salivary IGHG3, IGHA1, S100A8, lactoferrin, and OGG1 determined by Western blotting and the concentrations of salivary IGHG3 and lactoferrin measured by ELISA in patients with SLE or RA and HCs. As the ELISA data were normally distributed and the variance was homogeneous among the groups, a one-way analysis of variance was used. The correlations between the levels of salivary IGHG3 or lactoferrin and the clinical features in patients with SLE were determined using the Spearman’s rank correlation coefficient. Using the ROC curve of the salivary proteins, the AUC, sensitivity, and specificity were calculated. The difference was considered statistically significant when the *p*-value was less than 0.05. All statistical analyses were performed using the Statistical Package for the Social Sciences version 22.0 (IBS Corp, Armonk, NY, USA) and SAS 9.4 (SAS Institute Inc., Cary, NC, USA).

## Figures and Tables

**Figure 1 ijms-22-01374-f001:**
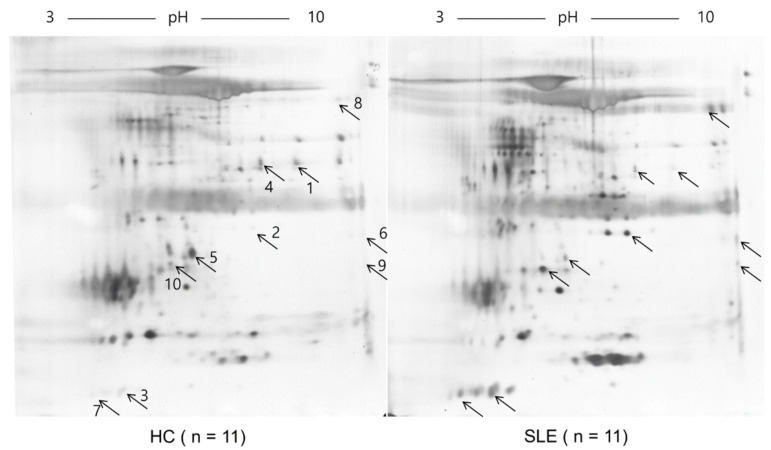
Representative two-dimensional gel electrophoresis protein map in systemic lupus erythematosus (SLE) and healthy controls (HCs). The salivary samples from 11 patients with SLE or 11 HCs were pooled equally, and isoelectric focusing was conducted with an immobilized pH gradient strip and an isoelectric point of 3–10 nonlinear. The concentration of 10 separated spots (marked as arrows) was different between the two groups. The experiments were performed in triplicate.

**Figure 2 ijms-22-01374-f002:**
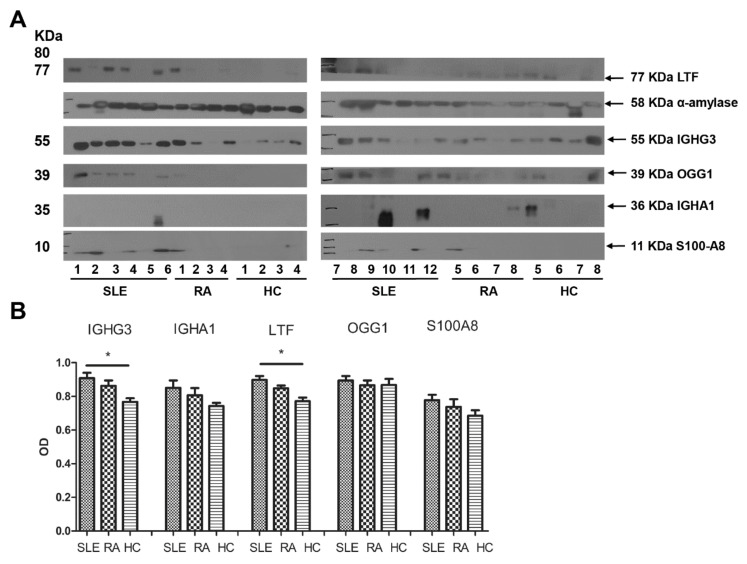
Expression levels of the salivary proteins in healthy controls (HCs) and patients with systemic lupus erythematosus (SLE) or rheumatoid arthritis (RA). A Western blot analysis was performed on 12 age-matched saliva samples of patients with SLE, 8 saliva samples of patients with RA, and 8 saliva samples of HCs. Patients with SLE exhibited an enhanced expression of salivary immunoglobulin gamma-3 chain C region (IGHG3) and lactoferrin (LTF) compared to HCs. OGG1, 8-oxoguanine DNA glycosylase; IGHA1, immunoglobulin alpha-1 chain C region; and S100A8, protein S100-A8. * *p*-value < 0.05.

**Figure 3 ijms-22-01374-f003:**
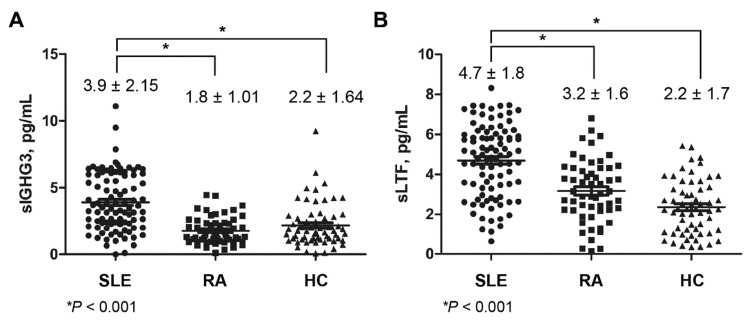
Concentrations of the salivary immunoglobulin gamma-3 chain C region (sIGHG3) (**A**) and salivary lactoferrin (sLTF) (**B**). The levels of sIGHG3 and sLTF were measured in patients with systemic lupus erythematosus (SLE) or rheumatoid arthritis (RA) and healthy controls (HCs) by an enzyme-linked immunosorbent analysis. The levels of salivary IGHG3 and lactoferrin were elevated in patients with SLE compared to those in HCs and patients with RA. All measurements were performed in duplicate. All values are presented as the mean ± standard deviation. * *p*-value < 0.001.

**Figure 4 ijms-22-01374-f004:**
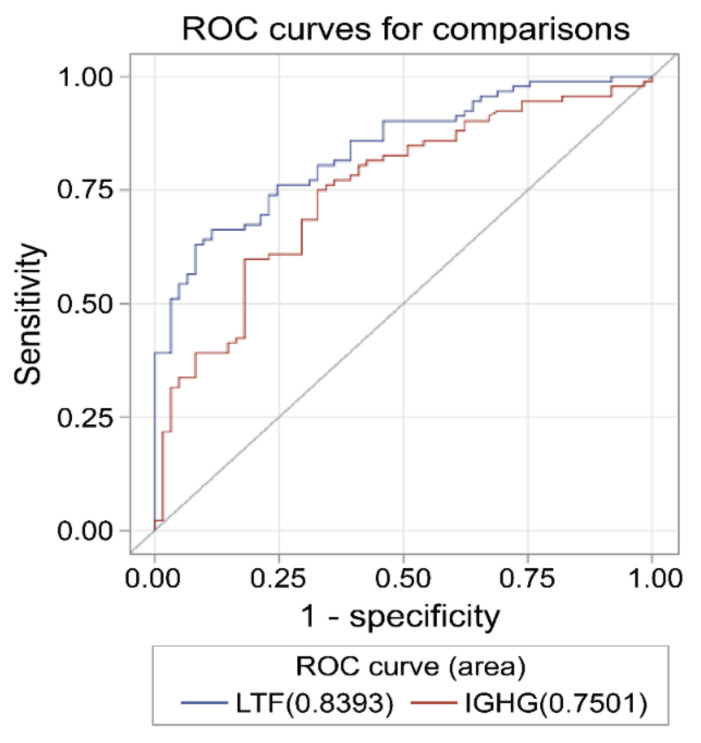
Receiver operating characteristic (ROC) curves for the salivary immunoglobulin gamma-3 chain C region (IGHG3) and lactoferrin (LTF) for the diagnosis of systemic lupus erythematosus (SLE). The area under the curve (AUC), sensitivity, and specificity of sIGHG3 were 0.75, 78%, and 64.6%, respectively, with a cut-off value of 2.26 pg/mL for the diagnosis of SLE compared to healthy controls (HCs). The AUC, sensitivity, and specificity of salivary LTF were 0.84, 91.9%, and 60.6%, respectively, with a cut-off value of 4.36 pg/mL for the diagnosis of SLE.

**Figure 5 ijms-22-01374-f005:**
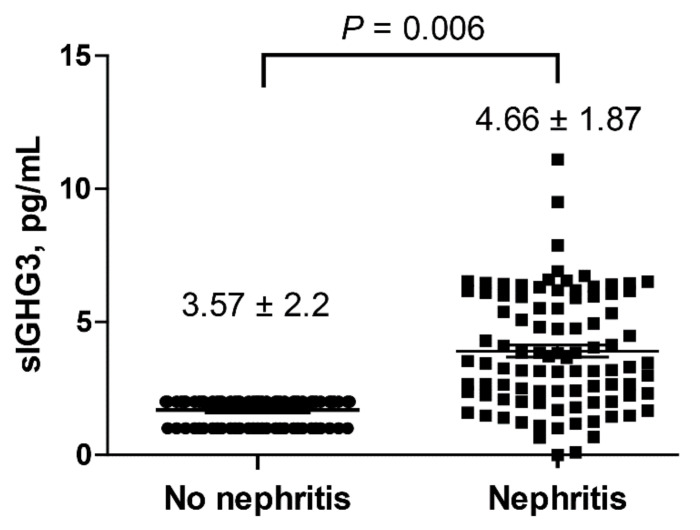
Concentrations of salivary immunoglobulin gamma-3 chain C region (sIGHG3) in patients with and without lupus nephritis (LN). The levels of sIGHG3 in patients with LN were significantly higher than those in patients without LN. All measurements were performed in duplicate. All values are presented as the mean ± standard deviation.

**Table 1 ijms-22-01374-t001:** List of salivary peptides with different concentrations of the 2D electrophoresis analysis between patients with systemic lupus erythematosus and health controls.

Spot	Increased Protein Name	Fold of Variation SLE vs. HC	ANOVA, *p*-Value
2	Immunoglobulin gamma-3 chain C region	4.101	0.000158
3	S100A8	3.894	0.0045
6	8-oxoguanine DNA glycosylase	3.516	0.00637
7	S100A8	2.947	2.48 × 10^−8^
8	Lactotransferrin	2.827	4.06 × 10^−7^
9	8-oxoguanine DNA glycosylase	2.794	0.00653
**Spot**	**Decreased Protein Name**	**Fold of Variation SLE vs. HC**	**ANOVA, *p*-Value**
1	Alpha-amylase 1	−4.228	0.000691
4	Alpha-amylase 1	−3.676	0.0372
5	Immunoglobulin alpha-1 chain C region	−3.532	0.00511
10	Leukemia-associated protein 7	−2.176	0.0178

SLE, systemic lupus erythematosus; HC, healthy control; and ANOVA, analysis of variance.

**Table 2 ijms-22-01374-t002:** Correlation of salivary proteins and clinical characteristics of systemic lupus erythematosus.

Clinical Characteristics	IGHG3, pg/mL		Lactoferrin, pg/mL	
	*r*	*p*-Value	*r*	*p*-Value
ESR	0.26	0.01	−0.09	0.4
Complement 3	−0.02	0.89	0.21	0.05
Complement 4	−0.02	0.88	0.2	0.06
Anti-dsDNA Ab (+)	0.25	0.01	−0.05	0.63
Mucocutaneous involvement	0.15	0.14	0.08	0.48
Arthritis	−0.1	0.36	0.01	0.9
Nephritis	0.28	0.01	−0.01	0.94
Serositis	0.04	0.74	−0.15	0.16
Hematologic disease	−0.02	0.83	−0.08	0.45
SLEDAI	0.03	0.74	−0.09	0.42
Use of glucocorticoids	0.15	0.06	−0.07	0.41
Dose of glucocorticoids	0.09	0.25	−0.05	0.57
Use of immunosuppressants	0.21	0.04	0.02	0.85

IGHG3, immunoglobulin gamma-3 chain C region; ESR, erythrocyte sedimentation rate; dsDNA, double-stranded DNA; Ab, antibody; and SLEDAI, systemic lupus erythematosus disease activity index.

**Table 3 ijms-22-01374-t003:** Clinical characteristics of the subjects.

Clinical Characteristics	SLE	RA	HC
Number	94	57	62
Age, years	39.8 ± 9.8	41.0 ± 7.9	39.5 ± 6.9
Sex (F/M)	87/7	50/7	58/4
Leukocyte, µL	5165.4 ± 2364.9		
Hemoglobin, µL	12.2 ± 2.4		
Platelet, ×10^3^, µL	221.8 ± 76.1		
Lymphocyte, µL	1454.0 ± 654.0		
ESR, mm/h	16.4 ± 18.0		
Complement 3, mg/dL	85.2 ± 26.9		
Complement 4, mg/dL	18.6 ± 8.9		
Anti-dsDNA Ab (+), *n* (%)	41 (43.6)		
Rheumatoid factor (+), *n* (%)		45 (77.6)	
Mucocutaneous involvement, *n* (%)	28 (29.8)		
Arthritis, *n* (%)	31 (33.0)		
Nephritis, *n* (%)	29 (30.9)		
Serositis, *n* (%)	4 (3.8)		
Hematologic involvement, *n* (%)	35 (37.6)		
SLEDAI	3.8 ± 4.2		
DAS-28		3.3 ± 1.15	
Use of glucocorticoids, *n* (%)	69 (73.4)	44 (77.2)	
Dose of glucocorticoids, mg *	3.1 ± 3.7	2.3 ± 1.8	
Use of immunosuppressants, *n* (%)	35 (37.2)		

All values presented as number (%) or mean ± standard deviation. SLE, systemic lupus erythematosus; RA, rheumatoid arthritis; HC, healthy control; ESR, erythrocyte sedimentation rate; dsDNA, double-stranded DNA; Ab, antibody; SLEDAI, systemic lupus erythematosus disease activity index; and DAS28, disease activity score including 28 joints. * Prednisolone-equivalent dose.

## Data Availability

Not applicable.

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
