# Peer review of "Salivary Immunoglobulin Gamma-3 Chain C Is a Promising Noninvasive Biomarker for Systemic Lupus Erythematosus"

_ijms, 2021, doi:10.3390/ijms22031374_

Round 1

Reviewer 1 Report

This study analysed the salivary protein components in SLE patients compared to healthy controls using 2D gel electrophoresis, mass spectrometry and western blot. The authors identified 7 molecules that were differently expressed be 2D GE, but western blot confirmed mostly IGHG3 and lactoferrin to be upregulated in saliva of SLE patients. Quantifying these molecules by ELISA, in particular IGHG3 was found to moderately correlate with parameters of disease activity in a larger cohort of SLE patients.

This is an interesting observation but insights into pathogenic mechanisms remain vague and the utility of the markers in the clinical routine seems to be limited.

Concerns:

  1. Introduction: The sentence on anti-dsDNA antibodies (line 37-38) should be referenced (original work), as the statement is not common knowledge. Ref 41 is a review and should be replaced by the original work, too. The activation of the complement system (lines 43-44) does not lead to deficiency but to consumption/depletion. The genetic variability of the concentrations of complement components with regard to lab parameters that are used in SLE is mostly described for C4 (but not necessarily for all components). The difficulties in using complement concentrations as parameters of disease activity are more complex than just genetic factors.
  2. Results: For 2D gels, samples of healthy controls and SLE patients were pooled. Thus, differences in spot intensities might be due to the overexpression of certain molecules in a few (maybe even one single) individuals. Although western blot data partially refute this hypothesis, this limitation is not well clarified. It would have been more convincing to use smaller pools of samples. The signal intensities of Western blots obviously have been quantified in order to allow comparisons between patient groups. This quantification is not shown/explained. In addition, the resulting differences are not intuitive when having a look at the raw data. For example, one might have expected differences for OGG1 as well (or even more than for IGHG3). Quantification data should be shown (maybe as a part B of the Figure). Furthermore, it remains unclear whether upregulation of IGHG3 and lactoferrin is an unspecific phenomenon (with upregulation of all kind of proteins) or whether it is specific. For example, what is happening when quantifying albumin or one of the complement molecules in saliva ? With regard to the ROC curves, it remains unclear against which population the SLE patients were compared (probably pooled HC/RA; only appropriate if there was no difference between these two control groups). Figures 3 and 5: The bars among the dots: median or mean ? (as data was not normally distributed, it would have to be the median (maybe with IQR))
  3. Discussion: In lines 163-164 the authors state that the 10 dots identified by 2D-GE were consisting of 6 molecules but according to table 1 there were 7 (?). The statement in line 170 that anti-dsDNA interfering with pleural mesothelial cells is THE pathogenic mechanism underlying serositis is not fully supported by the cited study in which mostly total IgG was studied and associations with anti-dsDNA were found. Similarly, the second last sentence (215-216) should use ‘may be used’ instead of ‘can be used’, as the diagnostic utility is rather limited. I had difficulties in getting the key messages of the second (168-185: What do the authors want to imply ?) as well as of the third paragraph (192-207: What is the hypothesis of the authors on lactoferrin incorporating the previous studies and their own data ?).

Reviewer 2 Report

This study performed salivary protein analysis on SLE patients, and identified candidate protein biomarkers, IGHG3 and lactoferrin. Their diagnostic impact is modest with ROC value of 0.79, but they could have some pathophysiological importance to SLE. However, a major revision is required to improve technical clarification and discussion of this article.

#major points#

#The description of study population is not sufficient. The demographics should be provided as separate tables. Were 11 lupus patients for 2D electrophoresis and 14 SLE patients for western blotting overlapping with Table 3 ELISA patients? Were they active patients? S100A8 is an abundant protein in neutrophils, and it could be related to neutrophil levels in blood. Importantly, the treatment information should also be provided. Were they treated with steroids or other immune suppressants?

#Figure 2. Not only the raw ELISA figures, but also the protein concentrations of IHGH3 and other proteins should be provided as boxplots or other plots.

#Table 2. The correlation rho values are low, which suggest the correlations are weak, even if significant. Did the treatment, such as steroid dose, affect the protein levels of IGHG3 and lactoferrin in saliva?

#The strength of this study is that it is focusing on saliva samples. This study is interesting, but if you could add blood tests of IGG3 levels of the same patients, it will reveal whether elevation of salivary IGHG3 is dependent of blood changes. Also, more discussion on the difference with blood IGG3 isotype use is needed. For example, Ken Smith group has performed blood B cell receptor repertoire analysis of untreated lupus patients, but IGG3 isotype use was not elevated in lupus, but in EGPA. Nature. 2019 Oct;574(7776):122-126.

#minor points#

#Figure 1. How many spots were identified in total?

#Table 1. Are these p-values corrected for multiple testing? Why ANOVA tests were used?

#Method 4.3. Please use “g” instead of “rpm” for reproducibility of centrifugation.

Round 2

Reviewer 1 Report

The paper is surely improved but there are still concerns that should be addressed:

  1. Figure 2B is surely helpful in interpreting the Western Blot data but the different column patterns have to be explained in a figure legend (or the figure needs to be built-up differently). In addition, bars or stars indicating the significant differences between SLE and HC (in line with those being explained in the text) would be helpful.
  2. The new/revised paragraph on optical densities remains unclear. The first sentence starts with “Optical densities (ODs) of salivary IGHG3 were 0.91 ± 0.11, 0.86 ± 0.09 (vs SLE, p =0.28), and 0.77 ± 0.06 (vs SLE, p = 0.004) a …”. I understand that there were three different ODs (0.91, 0.86 and 0.77), but it is unclear to what these ODs correspond. Probably SLE, RA and HC respectively (?). In this order ? The text should be more clear in this. Same for the remaining paragraph.
  3. The authors state that the distribution of data was normally distributed and therefore mean +/- SD were used in figures 3 and 5. The use of mean/SD should be mentioned somewhere, either in the methods or in the figure legends. It also needs to be clarified why in the methods it is written that “data were not normally distributed and the variance was not homogeneous among the groups …”. As a consequence, here the authors draw the conclusion that non-parametric tests have to be used (Mann Whitney and Spearman). Please stay consistent.
  4. The new parts in the discussion might benefit from correction by a native speaker. I still had difficulties to get all messages.

Reviewer 2 Report

Thank you for your prompt revision. 

Unfortunately, I noticed some technical problems that need to be addressed. 

I feel you are nicely replying to me, but the manuscript is not revised accordingly. In the revision of the manuscript please revise the manuscript itself, not only commenting to me (reviewer2). To clarify which point is revised and which is not, please provide the revised sentences of the manuscript in the point-by-point response letter, too.

example) 

#revision point from the reviewer

your reply..

page xx line yy "revised sentence" This part is needed, too. 

Of course, if you need to argue that my suggestion is not constructive and you don't revise accordingly, it is ok. But in this reply, you are nicely replying to me in the letter, such as "there was no significant correlation" etc, but you are not providing the data in the manuscript. It is a big problem for acceptance. 

Figure 2B The x-axis labels are missing. Could you calculate the concentration of the proteins by the the standard curve? Thank you for the analysis.  

#major points#

#The description of study population is not sufficient. The demographics should be provided as separate tables. Were 11 lupus patients for 2D electrophoresis and 14 SLE patients for western blotting overlapping with Table 3 ELISA patients? Were they active patients? S100A8 is an abundant protein in neutrophils, and it could be related to neutrophil levels in blood. Importantly, the treatment information should also be provided. Were they treated with steroids or other immune suppressants?

#Figure 2. Not only the raw ELISA figures, but also the protein concentrations of IHGH3 and other proteins should be provided as boxplots or other plots.

#Table 2. The correlation rho values are low, which suggest the correlations are weak, even if significant. Did the treatment, such as steroid dose, affect the protein levels of IGHG3 and lactoferrin in saliva?

#The strength of this study is that it is focusing on saliva samples. This study is interesting, but if you could add blood tests of IGG3 levels of the same patients, it will reveal whether elevation of salivary IGHG3 is dependent of blood changes. Also, more discussion on the difference with blood IGG3 isotype use is needed. For example, Ken Smith group has performed blood B cell receptor repertoire analysis of untreated lupus patients, but IGG3 isotype use was not elevated in lupus, but in EGPA. Nature. 2019 Oct;574(7776):122-126.

#minor points#

#Figure 1. How many spots were identified in total?

#Table 1. Are these p-values corrected for multiple testing? Why ANOVA tests were used?

#Method 4.3. Please use “g” instead of “rpm” for reproducibility of centrifugation.

Round 3

Reviewer 1 Report

My specific concerns have all been adressed.

Author Response

Thank you for your positive response.